A modularity analysis helps improving the structure of the International Code of Zoological Nomenclature

Vlachos Evangelos evlacho@mef.org.ar evlacho@gmail.com
CONICET and Museo Paleontologico Egidio Feruglio , Trelew , Chubut , Argentina
Lefkowitz Elliot
Electronic publication date: 2021 Feb 23
Publication date: 2021
Volume: 9
Electronic Location ID: e10815
Received 2020 Nov 16; Accepted 2020 Dec 30
Copyright: ©2021 Vlachos
Copyright year: 2021
Copyright holder: Vlachos
License: This is an open access article distributed under the terms of the Creative Commons Attribution License, which permits unrestricted use, distribution, reproduction and adaptation in any medium and for any purpose provided that it is properly attributed. For attribution, the original author(s), title, publication source (PeerJ) and either DOI or URL of the article must be cited.
License URL: https://creativecommons.org/licenses/by/4.0/

Keywords: Nomenclature, Taxonomy, Zoology, Taxon, Code, Network analysis, Modules

Funding: The authors received no funding for this work.

==============================
Background

In a recent work I transformed a complex and integrated text like the International Code of Zoological Nomenclature into a network of interconnected parts of text. This new approach allowed understanding that a continuous body of text cannot accurately reflect the true structure of the Code, and provided a scientific methodology to identify a priori parts that could be affected by future revisions. In this next step, I investigate further the structure of the Code, seeking to use the network in order to identify the various conceptual communities grouping the various articles and other text items of the Code.

Methods

Using the first version of the network of the Code, I perform a comprehensive modularity analysis in two rounds: the first round aims to identify the fewest and largest communities or modules for the entire network, whereas the second round identifies the sub-modules within each larger module. The potential conflicts between the current structure of the Code and the module composition are evaluated with a parcellation analysis.

Results

The optimal modularity search identified 10 different modules in the entire network of varying size (ranging from 75 to 200 nodes). Each module can be further divided into smaller modules, that all-together allow describing the 65 conceptual groups of text items in the Code. Parcellation analysis revealed that two-thirds of the current chapters of the Code are in excellent or good accordance with the recovered conceptual modules, whereas the current composition of six chapters is in serious conflict with the conceptual structure of the Code.

Discussion

Judging only the composition and not the order of appearance of the Articles in the Chapters of the Code, I show that in many cases the current structure of the Code is found to correspond quite well to the concepts presented therein. The most important conflict is found on the provisions related to the various groups of names governed by the Code: family-, genus-, and species-group names. Currently, these provisions are spread out in different Articles in different Chapters, along the entire length of the Code. The modularity analysis suggests that re-organizing the Code in chapters that will deal with all aspects related to a given group (e.g., chapters including information on name formation, availability, typification, and validity for a given group), could potentially improve reader experience and, consequently, the applicability of the Code.

Introduction

Zoological Nomenclature, as defined by the International Code of Zoological Nomenclature (henceforth: the Code) is governed by the International Commission on Zoological Nomenclature (henceforth: the Commission) (ICZN, 1999). Recently, I managed to combine Zoological Nomenclature with recent advances in network theory, and create a new perspective for the Code (Vlachos, 2019). I successfully transformed the Code into a network—called Neticon zoologicon—and started to understand how this scientific language functions. I showed how a network framework for the Code helps predicting parts that need improvement and parts that might be affected by upcoming revisions. This new framework can also allow comparing different Codes, both those that are truly different to each other (e.g., the Zoological and the Botanical Codes) or different editions of the same Code. The Neticon can also be used to present the Code in a different way, improving both reader experience and usage, as well as enhancing the teaching potential. This last point is fully investigated herein, with a modularity analysis of the Neticon.

Any network of interconnected nodes has a structure, as some of the nodes are more closely connected to each other than the rest of the network, forming clusters or communities called modules. The discovery of these modules is a principal objective for network science. In this case, the recognition of several modules in the Code could reveal the true hierarchical structure of the Code, and would allow to formulate hypotheses for future improvements and changes.

Methods

The starting point for this analysis is Neticon zoologicon v. 1.0, the network constructed and presented in Vlachos (2019); see therein for details regarding the network itself. The modularity analysis is based on the built-in Blondel et al. (2008) algorithm, that follows an heuristic approach, performed in the free software Gephi 0.9.2 (Bastian, Heymann & Jacomy, 2009). Starting from a random point in the network, the algorithm heuristically tries to identify communities based on a pre-defined resolution scale; herein, the maximum resolution (1.0) was selected, to allow finding the largest communities or modules. Because of its heuristic approach, each run of the algorithm may produce slightly different results in the composition of modules. This has been discussed extensively in the literature, and methods have been proposed to decide the minimum number of runs necessary to get the best results, which depends on the total number of nodes in the network (Calatayud et al., 2019 and references therein). Blondel et al.’s (2008) algorithm is not affected from the starting point and the final results show a variation in the scale of 10−2. The numbering of the modules is random and depends on the random starting point of the modularity algorithm.

The modularity analysis made herein comprises two rounds. In the first round, the entire network is analysed, running the algorithm with 1.0 resolution for 50 times, keeping the solution with the maximum Q. The second round is made on each of the large modules recovered during the first round and filtered as separate modules, with another 25 runs of the algorithm with the same settings and always maximizing the value Q. The scope behind this two-round search is to create an hierarchical structure of the conceptual communities of the Code. It must be noted here that the current, de facto, hierarchical structure of the Code—the order and composition of Chapters and their containing Articles—is also coded in the Neticon, however with edges of a minimum weight compared to the conceptual ones (see Vlachos, 2019 for more information). The used algorithm takes into account both the weight and the direction of the edges. The decision to include in the modularity analysis these structural connections is made because I would like to propose the most conservative changes in the de facto structure of the Code. Another point that should be noted here is that the composition of the recovered modules is not to be taken literally, and especially for the nodes placed in the periphery of each module. These nodes are, actually, in the contact zone between two or more modules and—especially in the cases where these nodes are equally connected with both modules—most times their placement in one or the other module changed during each run. Therefore, some of the “solitary” items that might appear in some modules could actually equally be placed in another module. What I am interested here is the main and general composition of each module.

The comparison between the recovered modules is made with a modified Parcellation Index (PI) per Chapter and Article, originally developed for Anatomical Networks in Esteve-Altava et al. (2019). Once the analyses are made, in each Chapter/Article I count the number of items include in each recovered module. For a network of N nodes and 3 modules, for example, the PI would be calculated as such: PI = 1 −[(N1/N)2+(N2/N)2+(N3/N)2] (Esteve-Altava et al., 2019); this metric takes into account not only how many modules are present, but also the number of Nodes in each module. A Chapter/Article that has all its items recovered in a single module would have a PI = 0, whereas a Chapter/Article that an equal number of nodes placed in all recovered modules would have a PI = 1. Ideally, the de facto structure should match the conceptual structure, i.e., each chapter should be included in one module and thus having a PI close or equal to zero. On the contrary, if a chapter is included in many different modules, it means that its composition is in conflict with the conceptual structure of the Code. The network allows describing precisely this issue, by calculating the Chapter and Article individual parcellation index. If all the parts of one Chapter are included in a single module, then its parcellation index is 0; this means that the de facto structure of these Chapters is in harmony with the conceptual structure of the Code. This technique helps evaluating the structure of the various Chapters of the Code. We could divide, arbitrarily, the parcellation spectrum in four equal areas: those Chapters with PI between 0–0,250 have excellent structure, taking into account the issue with the solitary items explained above, that could slightly increase the PI; Chapters with PI between 0,250–0,500 have good structure, as the great majority of their items is included in one Module; Chapters with PI between 0,500–0,750 have problematic structure, as their items are spread across many Modules; Chapters with PI between 0,750–1,000 have bad structure, being parcellated across nearly all modules and with several items included in each one. This evaluation can also help improving the reader experience of the Code. A Chapter with low PI is a Chapter that can be read separately, whereas a Chapter with high PI requires reading it within the entire context of the Code.

Results

The optimal modularity search (run #23/50) discovered 10 different modules (module 0 –module 9; keeping the original numbering of the software), with a Q of 0,586 (standard deviation= 0,004 among the 50 runs, which is within the range reported by Blondel et al. (2008) and a high parcellation index of 0,889 (Table 1 and Supplemental Information). The modularity analysis reveals that the Code has a highly modular structure, meaning that it could be divided in at least 10 distinct and large groups that contain text items that are more closely connected to each other compared to the rest of the Code. This means that exist at least 10 different conceptual and general themes developed across the text of the Code. The high general parcellation index corroborates the highly modular structure of the Code. This number is to be interpreted as not only having many modules, but mainly that each one of these modules is important for the entire structure.

Table 1 Distribution of the items of each chapter of the code across modules.

The various text parts of the Chapters (Ch.) of the Code are included in different conceptual modules, as revealed by the modularity analysis. “Other” refers to texts not included in the Chapters, i.e., the Glossary and the Appendices. Their Parcellation Index (PI) is indicated as well.

	Modules		
Ch.	0	1	2	3	4	5	6	7	8	9	N	PI	
1	19	1	2	–	–	–	–	1	–	–	23	0,306	
2	0	14	–	–	1	–	–	–	–	–	15	0,124	
3	51	–	–	–	–	–	–	–	1	–	52	0,038	
4	6	41	41	2	9	2	11	11	10	1	134	0,787	
5	27	–	1	–	–	3	–	2	–	–	33	0,318	
6	1	3	1	47	1	–	4	1	–	–	58	0,335	
7	–	25	–	1	39	35	27	–	1	–	128	0,75	
8	2	–	–	4	2	–	21	–	–	–	29	0,447	
9	–	1	–	1	1	–	5	7	–	–	15	0,658	
10	–	16	1	2	–	1	5	2	1	–	28	0,628	
11	3	4	10	2	3	13	–	4	–	1	40	0,798	
12	1	2	1	72	2	–	5	2	1	3	89	0,339	
13	–	–	–	2	–	–	2	2	10	–	16	0,563	
14	–	–	–	–	–	–	11	–	–	–	11	0	
15	–	2	–	3	–	–	1	85	–	–	91	0,126	
16	–	1	1	1	–	–	1	–	135	1	140	0,07	
17	1	–	–	9	–	–	2	–	2	80	94	0,266	
18	–	–	–	–	–	–	–	–	–	16	16	0	
Other	70	43	22	41	24	21	43	30	39	34	367	0,306	
N	181	153	80	187	82	75	138	147	200	136	1379	0,889	

Figure 1 The main modules of the Code.

The 10 different conceptual modules of the International Code of Zoological Nomenclature based on the modularity analysis of the network of the Code (Neticon zoologicon). The modules are depicted in a strong gravity Force Atlas 2 layout. The most important nodes are highlighted: AUT, author; ART, Article; AVN, available name; COD, Code; COM, Commission; FAN, family name; GEN, genus; PUB, Publication; REC, recommendation; SP, species; SPN, species name; TAX, taxon; TSP, type species; TY, type; VA, valid name.

Module 0 contains 181 Nodes and 833 Edges, representing 13,13% of the Code; it is the third-largest module in the network (Fig. 1). Module 0 mostly contains items from Chapter 1 (Zoological Nomenclature; most items of Art. 1, and the entire Arts. 2, 3), Chapter 3 (Criteria of Publication; nearly all items, excluding Rec. 8B which forms part of Module 8), some items from Chapter 4 (Criteria of Availability; parts of Art. 10 and Art. 17), most of Chapter 5 (Date of Publication; the entire Art. 21, and some parts of Art. 22), as well as some parts from Chapters 6 (Rec. 24A), 8 (Art. 40.2.1, and Rec. 40A), 11 (Recs. 50B, 50C, 51F), 12 (Art. 54.1), and 17 (Art. 78.2.4). The most important Glossary terms included in this Module are: publish, date of publication, scientific name, publication, nomenclatural act, and zoological nomenclature. This Module also contains the Code of Ethics (Appendix A of the ICZN) and General Recommendations (Appendix B of the ICZN). Therefore, Module 0 covers some Generalities and introductory information of the Code, as well as what constitutes published and unpublished work. A secondary modularity analysis allows understanding better the conceptual structure of Module 0. It consists of nine sub-modules, which deal with the following topics (Fig. 2): sub-module 0.0: Introduction and Generalities and what is excluded from the Code and articles on the availability of excluded names (Arts. 1–3, Art. 10.5, Arts. 17.1–3, Rec. 50C, Rec. 51F, Art. 54.1); sub-module 0.1: what constitutes published work, excluding electronic publication (almost the entire Art. 8, excl. the sub-articles of 8.4 and 8.5); sub-module 0.2: what does not constitute published work (Art. 9); sub-module 0.3: date of publication (Art. 21); sub-module 0.4: citation of date (Art. 22 and Art. 40.2.1); sub-module 0.5: electronic publication and generalities on publication (Art. 8.5, and various recommendations related to publication); sub-module 0.6: publication as physical copies (Art. 8.4); sub-module 0.7: General Recommendations (3–9); and sub-module 0.8: Ethics and Recommendations (Code of Ethics and General Recommendations 10–12).

Figure 2 The composition of the different modules of the Code.

Each of the 10 main modules of the Code (see Fig. 1) deals with important concepts, and contains several smaller sub-modules that group together the provisions on various smaller conceptual elements. The Code can be divided into 65 different conceptual elements, grouped into 10 larger ones, indicating the conceptual structure of the Code.

Module 1 contains 153 Nodes and 748 Edges, representing 11,09% of the Code; it is the fourth-largest module in the network (Fig. 1). Module 1 mostly contains items from Chapter 2 (Number of words in scientific names; all items except Art. 4.1), a large part of Chapter 4 (Criteria of Availability; most of Art. 11 and parts of Arts. 10, 15, and 16), a large part of Chapter 7 (Formation and treatment of names; mostly those parts that deal with the spellings of specific names), the largest part of Chapter 10 (Species-group nominal taxa and their names), and the part of Chapter 11 dealing with species-group names (Authorship; Art. 50.3.1, Rec. 51A, Art. 51.3.3 and its example). This Module also contains a few individual items from Chapter 1 (Art. 1.3.4), Chapter 6 (Arts. 23.3.3, 23.3.4, and Example of Art. 24.1), Chapter 9 (Art. 42.1), Chapter 12 (some examples and recommendations on specific names), Chapter 15 (Art. 67.5.3 and the Example of Art. 67.12.1), and Chapter 16 (Rec. 72C). The most important Glossary terms included in this Module are species-group name, species group, name of species, combination, specific name, binomen, subspecies, and other terms related to the species group. Therefore, Module 1 deals primarily with names in the species group, their availability, as well as with general concepts like the binominal nomenclature. It should be noted that articles dealing with types in the species groups are not included in this Module, but are mostly in the adjacent Module 8 (see below). A secondary modularity analysis allows understanding better the conceptual structure of Module 1. It consists of six sub-modules, which deal with the following topics (Fig. 2): sub-module 1.0: scientific names and binomina (most of Arts. 4–6, and other related items); sub-module 1.1: species-group names (Art. 45, and other related items); sub-module 1.2: availability of names of collective groups, ichnotaxa, divisions and other special cases (Arts. 10.3, 10.4, 10.7); sub-module 1.3: availability requirements (most of Art. 11); sub-module 1.4: availability of species-group names (Art. 11.9.3); sub-module 1.5: species-group names and spellings (Arts. 31, 32 and other related items).

Module 2 contains 80 Nodes and 418 Edges, representing 5,8% of the Code; it is the second smallest module in the Network (Fig. 1). Module 2 mostly contains items from Chapter 4 (the majority of the information contained in the Articles in this chapter, exc. Art. 11 that is contained in other modules) and Chapter 11 (most of Art. 50, on authors). This Module also contains some individual items from other Chapters, including Chapter 1 (Arts. 1.3.6, 1.3.7), Chapter 5 (Rec. 22A.1, on the citation of date), Chapter 6 (Art. 23.3.7), Chapter 10 (Art. 45.5.1), Chapter 12 (Art. 54.2), and Chapter 16 (Art. 72.5.1) . The most important Glossary terms included in this Module are: available name, plenary power, Articles, taxonomic information, and unavailable name. Therefore, this Module deals primarily with the concept of availability of names named before or after some special dates, but without the articles that deal with publication (that are mostly included in Module 0). There is a significant overlap and contact between Module 2 and 0, which together contain most information on the availability of names. A secondary modularity analysis allows understanding better the conceptual structure of Module 2. It consists of five sub-modules, which deal with the following topics (Fig. 2): sub-module 2.0: citation of authors (Rec. 22A.1, Art. 50.2); sub-module 2.1: availability before 1931 (Art. 12.2 and related items); sub-module 2.2: availability after 1930 (Art. 13 and related items); sub-module 2.3: additional provisions on availability (the second half of Chapter 4 and related items); sub-module 2.4: additional provisions for the citation of authors (Arts. 50.1.1, 50.1.3).

Module 3 contains 187 Nodes and 993 Edges, representing 13.56% of the Code; it is the second largest Module in the network (Fig. 1). Module 3 mostly contains items from Chapter 6 (Validity; most items of Arts. 23 and 24), Chapter 12 (Homonymy; entire Arts. 52, 56, and most of the other Articles of that Chapter), and several Articles from Chapter 17 (Commission; Arts 81.2, 81.2.1–4, 82, 82.1, 82.2). This Module also contains some individual items from other Chapters, including Chapter 4 (examples of Arts. 11.5.2 and 11.6.1), Chapter 7 (example of Art. 33.3.1), Chapter 8 (Arts. 38, 39, and examples of 40.1 and 40A), Chapter 9 (Art. 44.2), Chapter 10 (Arts. 44.2 and 47.2), Chapter 11 (Art. 50.6 and its example), Chapter 13 (Art. 61.2.1 and its example), Chapter 15 (Art. 67.1.2, Rec. 67B, and the example of Art. 67.8.1), and Chapter 16 (Art. 72.9). The most important Glossary terms included in this Module are: valid, species homonym, valid name, synonym, family homonym, genus homonym, Principle of Priority, and priority. This Module also contains the most important Article of the Code, which is Art. 23.1. Clearly, Module 3 deals with the validity of names, expanding also to the concepts of homonymy and synonymy which are tightly connected. A secondary modularity analysis allows understanding better the conceptual structure of Module 3. It consists of eight sub-modules, which deal with the following topics (Fig. 2): sub-module 3.0: priority and homonymy of genus-group names (most of Art. 23, excl. provisions on family-group names and reversal of precedence, and Art. 56); sub-module 3.1: reversal of precedence (Art. 23.9–10, and related items); sub-module 3.2: precedence of simultaneously published names, spellings or acts and the First Reviser (Art. 24 and related items); sub-module 3.3: homonymy in the family group and their suppression (Arts. 38, 39, 54, 55, 81.2, and related items); sub-module 3.4: additional provisions on validity, homonymy, synonymy (Arts. 23.3, 23.7, 23.12, 60.1–3, and related items); sub-module 3.5: Principle of Homonymy (Art. 52 and some related items); sub-module 3.6: homonymy in family, genus, species, and variant spellings; and sub-module 3.7: secondary homonyms (Arts. 57.3, 59, and related items).

Module 4 contains 82 Nodes and 351 Edges, representing 5.95% of the Code; it is the third smallest module of the network (Fig. 1). Module 4 mostly contains the majority of the items from Chapter 7. Exceptions include mostly Arts. 29 (Family-group names) and 30 (Gender of genus-group names) which are included in Modules 6 and 5 respectively. Module 4 also contains items from Chapter 4 (Arts. 11.9.1.1–4 on species-group names, Arts. 19.1–4 on spellings), Chapter 6 (Art. 24.2.3 on original spellings), Chapter 8 (Art. 35.4 and the example of Art. 35.4.1), Chapter 9 (Art. 42.4), Chapter 11 (Arts. 50.4, 50.5, 51.3.1, on authorships), and Chapter 12 (Art. 54.3 and 58.14, on spellings). The most important Glossary terms are: spelling, new scientific name, incorrect subsequent spelling, incorrect original spelling, and original spelling. Therefore, this Module contains most of the information on the formation of names and spellings, focusing more on species-group names and excluding family- and genus-group names. A secondary modularity analysis allows understanding better the conceptual structure of Module 4. It consists of five sub-modules, which deal with the following topics (Fig. 2): sub-module 4.0: formation of species-group names (Arts. 11.9, 31.1, 58.14, and related items); sub-module 4.1: spellings (Arts. 32, 32, and related items); sub-module 4.2: inadvertent errors (Art. 32.5.1, and related items); sub-module 4.3: emendations and their authors (Arts. 19.1–4, 32.2.2, 33.2.1–3, 50.4, 50.5, 51.3.1 and related items); sub-module 4.4: formation of names (Arts. 25–34, and related articles).

Module 5 contains 75 Nodes and 281 Edges, representing 5.44% of the Code; it is the smallest module of the network (Fig. 1). Module 5 mostly contains items from Chapter 7, and especially those from Art. 30 on the Gender of genus-group names. It also contains many items from Chapter 11, that is Articles dealing with the authorship of names and especially in combination with genus-group names. Additionally, this Module contains some items from Chapter 4 (example of Art. 11.3 and Rec. 11A), Chapter 5 (Art. 22 and Recs. 22A and 22A.3, dealing with the citation of date and especially in combination), and Chapter 10 (Art. 48, on change of generic assignment). The most important Glossary terms are: author, Latin, gender, gender ending, Greek, compound, latinize, gender agreement, and new combination. Therefore, this module deals also with the formation of names, focusing however in the genus-group name, combination and gender agreement in particular. A secondary modularity analysis allows understanding better the conceptual structure of Module 5. It consists of five sub-modules, which deal with the following topics (Fig. 2): sub-module 5.0: gender of names formed from Latin or Greek words (Art. 30.1); sub-module 5.1: gender of names not formed from Latin or Greek words (Art. 30.2); sub-module 5.2: gender agreement (Arts. 31.2, 34.2, 48); sub-module 5.3: citation of author and date, related mostly to combinations (Arts. 22, 50.3, 51); sub-module 5.4: examples of the gender of names formed from Latin or Greek words (examples of relevant articles).

Module 6 contains 138 Nodes and 703 Edges, representing 10.01% of the Code; it is the sixth-largest module in the network (Fig. 1). Module 6 mostly contains items from Chapter 7, and especially those dealing with the formation of family-group names. It also contains several items from Chapter 4 (articles and examples on family-group names), most items from Chapter 8 (all on the family-group names), and Chapter 14 (all on name-bearing types of family-group names). Additionally, this module contains some individual items from Chapter 6 (priority on family-group names), Chapter 9 (Principle of Coordination in the genus-group), Chapter 10 (Principle of Coordination in the species-group), Chapter 12 (on homonymy in the family-group), Chapter 13 (types on nominotypical taxa), Chapter 15 (on type species), Chapter 16 (on type specimens), and Chapter 17 (on the use of plenary power related to rank). The most important Glossary terms included in this module are: taxon, family-group, family-group name, rank, scientific name, suffix, type genus, stem, family name, superfamily, Principle of Coordination, subfamily name, and superfamily name. Therefore, this Module mostly deals with the formation and application of family-group names, and its name-bearing types. This Module also contains the basic information on the Principle of Coordination and, thus, shows a strong overlap with Modules 7 and 8 that deals with the genus- and species-group in particular. A secondary modularity analysis allows understanding better the conceptual structure of Module 6. It consists of six sub-modules, which deal with the following topics (Fig. 2): sub-module 6.0: availability of family-group names (Art. 11.7.1, 64); sub-module 6.1: Principle of Coordination and some provisions on name-bearing types (Arts. 34.1, 36, 37, 43, 44, 46, 47, 61.2, 63.1, 67.1.1, 72.8); sub-module 6.2: formation of family-group names (Arts. 29.2, 29.4.1, 32.5.3, 35.2, 53.1); sub-module 6.3: formation of family-group names, stem and spellings (Arts. 23.1.2, 23.5, 29.1, 29.3–6, 33.2.3.1, 33.3.1, 35.4.1–2, 55.3.1 and related items); sub-module 6.4: validity and synonymy of family-group names (Arts. 23.1.1, 35.3, 35.5, 40.1, 40.2, 41, 63, 65); sub-module 6.5: miscellaneous information on family-group names, examples and definitions (examples of relevant articles and Glossary terms).

Module 7 contains 147 Nodes and 940 Edges, representing 10.66% of the Code; it is the fifth largest module in the network (Fig. 1). This Module contains the great majority of items included in Chapter 15 (types in the genus group) and several items from various Articles of Chapter 4 (all dealing with the genus-group names) and Chapter 9 (Genus-group). Additionally, this Module contains some individual items of Chapter 1 (Art. 1.2.1, on collective groups), Chapter 5 (examples on citation of dates related to genus-group names), Chapter 6 (Art. 23.7.2, priority in genus/collective group), Chapter 10 (Art. 49), Chapter 11 (some items on type species), Chapter 12 (some items on homonymy), and Chapter 13 (Art. 61.4 and its example, on nominal subgenus). The most important Glossary terms included in this module are: species, establish, genus group, genus-group name, type species, generic name, deem, and subgenus. Therefore, this module deals mainly with genus-group names and their application, as well as the type species. A secondary modularity analysis allows understanding better the conceptual structure of Module 7. It consists of six sub-modules, which deal with the following topics (Fig. 2): sub-module 7.0: type species (Art. 67); sub-module 7.1: availability of genus-group names and type fixation (Arts. 13.3, 16.3, 42, 66, 67.4.1, 67.12.1, 68, 69.4, 70.2, 70.3); sub-module 7.2: type species designations (Arts. 67.2, 67.3.2, 67.8.1, 69.1, 69.2); sub-module 7.3: subsequent designation recommendations (recommendations of Art. 69); sub-module 7.4: types in the genus group (several items of Arts. 67, 68, 69, 70 and related items); sub-module 7.5: type fixation, additional provisions (Arts. 67.2.1, 67.13, 68.2, 68.6, 70.3, 70.4 and related items).

Module 8 contains 200 Nodes and 978 Edges, representing 14.5% of the Code; it is the largest module in the network (Fig. 1). This Module contains the great majority of items included in Chapter 16 (Types in the species group), and several items from Chapter 4 (most of Art. 16 on names published after 1999) and Chapter 13 (most of Art. 61 on the Principle of Typification). Additionally, this Module contains some individual items of Chapter 3 (Rec. 8B), Chapter 7 (Rec. 25C), Chapter 12 (Art. 45.3), and Chapter 17 (Arts. 79.1.3 and 79.5.2). The most important Glossary terms included in this Module are: Recommendation, name-bearing type, specimen, nominal taxon, lectotype, holotype, syntype, neotype, description, type series, nomen novum, and taxonomy. Clearly, this Module deals with typification and particularly with types on the species group. A secondary modularity analysis allows understanding better the conceptual structure of Module 8. It consists of eight sub-modules, which deal with the following topics (Fig. 2): sub-module 8.0: Principle of Typification (Arts. 61.1, 61.3, and related terms); sub-module 8.1: types in the species group and neotypes (Art. 71, 72, 75, and related terms); sub-module 8.2: availability-related provisions on name-bearing types (Arts. 16.1, 16.2, 16.4, and related items); sub-module 8.3: recommendations on type specimens and type localities (recommendations of Arts. 16, 25, 72, 73, 74, 76); sub-module 8.4: type series (Art. 72.4 and related items); sub-module 8.5: holotypes, hapantotypes (Arts. 72.5, 72.6, 73.1, 73.3); sub-module 8.6: syntypes, lectotypes (Arts. 73.2, 74, and related items); sub-module 8.7: neotypes and type locality (Arts. 75.3, 76.1–3).

Finally, Module 9 contains 136 Nodes and 567 Edges, representing 9.86% of the Code; it is the fourth smallest module in the network (Fig. 1). This Module contains the great majority of items included in Chapter 17 (Commission), and all items of Chapter 18 (Regulations). Also, this Module contains individual items of Chapter 4 (Art. 10.6), Chapter 11 (example of 51F), Chapter 12 (Arts. 57.2.2–3, 59.3.1), and Chapter 16 (example of Art. 75.6). The most important Glossary items included in this Module are: Commission, Code, provisions, adopt, Opinion, ruling of the Commission, List of Available Names, Part of the List of Available Names, proposal, and index. Therefore, this module clearly contains almost all the information on the Commission and its regulations. A secondary modularity analysis allows understanding better the conceptual structure of Module 9. It consists of seven sub-modules, which deal with the following topics (Fig. 2): sub-module 9.0: Relations of the Commission (Art. 77, and related items); sub-module 9.1: powers and duties (Art. 78); sub-module 9.2: List of Available Names (Art. 79, and related items); sub-module 9.3: List of Available Names requirements (Art. 79.2); sub-module 9.4: Status of works, names and nomenclatural acts in Official Lists (Art. 80.6); sub-module 9.5: Opinions and Corrections (Arts. 80.2–5, 80.7, 80.9, and related items); sub-article 9.6: regulations governing the Code (Arts. 85–89).

Parcellation index

On one hand, the Code has a given, de facto, structure: it is the current structure of the Code, with its Articles that are grouped in Chapters, all placed in their order. On the other hand, the Code has another, conceptual, structure: this is the structure that is revealed by the modularity analysis and is based on the conceptual connections between the various parts of the Code. The PI allows calculating the agreement between these two structures (Table 1 and Fig. 3), and highlight how the presentation of the Code could be improved. When these two structures agree, the PI is low or zero. For example, Chapter 14 is included entirely in Module 6, and Chapter 18 is included entirely in Module 9. On the other hand, a Chapter that is spread across many Modules, with various items in each Module, will be highly parcellated, and its PI will be high, approaching 1. For example, the contents of Chapter 4 are placed in all 9 recovered Modules, and its parcellation index is 0.787 (Table 1). This new analysis reveals that six Chapters of the Code have excellent structure and composition (Fig. 3): Chapters 2, 3, 14, 15, 16, and 18. Another six Chapters have good structure and composition: Chapters 1, 5, 6, 8, 12, and 17. Therefore, two-thirds of the Code have actually a quite good conceptual structure that agrees quite well with the de facto structure. However, the remaining six chapters (4, 7, 9, 10, 11, and 13) are in the “red zone”, with two of them (4 and 11) crossing the 0.750 boundary. If we want to improve the current structure of the Code, we should focus on the placement and composition of these six problematic chapters.

Figure 3 Evaluation of the structure of the various Chapters of the Code.

Based on the recovered modules, the structure of each Chapter has been evaluated based on a parcelation analysis. Two-thirds of the Code appear to be included in Chapters with excellent or Code structure, whereas six chapters are considered to have problematic or bad structure. The most important problem is the inclusion of provisions dealing with names and taxa of different rank (species, genus, family) in the same Chapter.

Chapter 4 deals with availability and is arguably one of the most important Chapters of the Code. It contains many Articles (Arts. 10–20), matched only by Chapter 7 in number of Articles—it has, however, more individual items (134) toped only by Chapter 16 with 140 items. More importantly, the included Articles in this Chapter are quite heterogeneous and for three good reasons. First, Art. 10 contains few items, but most of these are related with parts contained in the first Chapters of the Code. Second, Chapter 4 includes items on the availability on names in different groups: family-group, genus-group, and species-group (Art. 11); these items are conceptually closer to the recovered modules dealing with these groups. Finally, this Chapter contains separate Articles that deal with the availability of names published in different times (1930, 1960, 1999), all of which are conceptually closer with other parts of the Code. Chapter 7 deals with the Formation and treatment of names, and suffers as well from one of the problems of Chapter 4: Art. 29 deals with family-group names, Art. 30 with genus-group names, and Art. 31 with species-group names. However, only the latter (Art. 31) appears to be highly parcellated. Also, the remaining Articles on spellings and emendations (Arts. 32–34) are also highly parcellated, as they also deal with names for taxa in different ranks. Chapter 13 deals with the Principle of Typification and suffers again from the same problem: its items are conceptually closer to modules dealing with types in different groups. The analysis of the PI of these three chapters reveals the first important problem of the structure of the Code: information dealing with names in the family, genus, and species group are placed in the same Article or Chapter, although they are conceptually different. A possible improvement, therefore, would be to create chapters the contain information for the same group only.

Chapter 9 deals with genus-group nominal taxa and their names. Although it contains three different Articles (Arts. 42–44), this is actually a quite short Chapter that contains only 15 items. Most of these are placed in Module 6 and 7, but some solitary items are placed elsewhere. Therefore the high PI is an artefact of the low number of items of this chapter, and its structure is not considered to be a significant problem herein.

Chapter 10 deals with the species group. Most of its items are placed in Module 1, however items from Arts. 46 and 47 that deal with Coordination are placed elsewhere. The parcellation analysis of these two chapters reveals another problem: the Principle of Coordination is an important concept that forms part of a different conceptual Module and it would be perhaps better to group all these articles together.

Finally, Chapter 11 deals with authorship, and is a highly parcellated chapter that contains heterogeneous information. For example, Art. 50 contains items that deal with many different concepts: authorship (Arts. 50.1–2), rank (Art. 50.3), combination (Art. 50.3), emendations (Arts. 50.4–5), date of publication (Art. 50.6), and synonymy (Art. 50.7). All these items are better grouped and placed elsewhere, in their conceptual modules, and that is why this Article is the one with the highest PI in the Code.

Contact between the modules

As we have seen, the contact between the various modules is quite extensive. All the modules together sum up to 8,928 Edges, meaning that 2,348 (a bit more than 20%) of the total 11,276 Edges of the network are responsible for connecting the modules to each other. The Neticon allows identifying exactly which are these Edges connecting Nodes between the modules (Table 2), and could be used for the fine-tuning of the structure of the Code. The most extensive contact zones are found between module 1 and module 7 (availability of species-group names and types in the genus group; 233 connecting edges), module 1 and module 3 (availability of species-group names and validity, homonymy, synonymy; 209 connecting edges), and module 0 and module 9 (introduction, generalities, publication and Commission, regulations; 197 connecting edges). Important contact zones are found between module 3 (validity, homonymy, synonymy) with module 6 (formation of names in the family group; 183 interconnecting edges), module 7 (types in the genus group; 186 interconnecting edges), and module 9 (Commission, regulations; 121 interconnecting edges), and the main availability module (module 1) with nearly all other modules. The least extensive contact between modules is found between module 4 and module 8 (formation of names in the species group and types in the species group; 10 interconnecting edges), and most other modules (exc. 0 and 3) with the module related to the Commission and regulations (module 9).

Table 2 Contact between the various modules of the Code.

The number of edges or connections between the various recovered modules of the Code.

Modules	0	1	2	3	4	5	6	7	8	9	
0	–	132	164	122	45	55	74	79	138	197	
1	132	–	140	209	105	127	127	233	163	45	
2	164	140	–	105	62	45	93	123	107	85	
3	122	209	105	–	68	29	183	186	94	123	
4	45	105	62	68	–	54	96	36	10	21	
5	55	127	45	29	54	–	63	74	66	18	
6	74	127	93	183	96	63	–	146	129	38	
7	79	233	123	186	36	74	146	–	147	41	
8	138	163	107	94	10	66	129	147	–	67	
9	197	45	85	123	21	18	38	41	67	–	

Discussion

The modularity analysis performed herein helps formulating recommendations for developing a more “reader-friendly” version of the Code, a version that will reflect more the conceptual clusters within the Code and less the historical, de facto, structure. This new version should be complementary to the current one, and the current numbering should be used as reference only and not for organization purposes. Having a version that is better organized could certainly improve the applicability of the Code, especially by those users that are not so experienced with all the Rules. Also, a better organized version would help a lot in teaching the Code to students and beginners.

The most important outcome of this analysis is that the greatest conceptual conflict is created by the placement of provisions dealing with the different and important concepts of the Code (publication, availability, typification, validity) for each group (i.e., family, genus, species group) in different chapters organized per concept. And this is further complicated if we add a temporal element, as there are in many cases different provisions for taxa named before or after certain dates. It seems that a different organization, grouping together all the provisions dealing with taxa and their names in the same rank, is desirable to improve the structure of the Code. The only concept that appears, currently, to override this organization is Homonymy, whose relevant provisions are mostly found in the same conceptual module.

Once this conflict is solved, the next step would be to resolve the second point of conflict, that is the different provisions depending on several Important Dates in the history of the Code. For the moment it seems that all relevant provisions for most of the specific dates form different sub-modules within the larger ones, so a new modularity analysis could help to prove whether more changes, and which, are necessary or not.

Also, the modularity analysis suggests that it might be desirable to merge Chapter 3 with Chapter 5. As we have seen, the items of Chapter 5 (Arts. 21 and 22) that deal with the date of publication and its form of citation, are conceptually included in the Module that also includes Chapters 1 and 3. Moving Chapter 5 before Chapter 4, or even include its Articles in Chapter 3, would agree more with the conceptual structure of the Code. Additionally, individual articles that are spread in other parts of the Code but also deal with the date of publication (e.g., Art. 40.2.1) could also merit inclusion. Chapter 11, on authorship, is highly parcellated, and it is better if it is dissolved and its articles placed in their corresponding modules.

Once the main modularity conflicts are resolved, another issue needs to be further investigated: the order of the various concepts in the continuous text. Currently, the order of appearance is from the higher rank to the lower one (i.e., family, genus, species). However, most of the modularity searches, following the connections within the Code, usually followed the inverse path: species, genus, family. This is something that needs to be further investigated, but it could perhaps provide an even better reader experience. Finally, in all runs, the beginning (Chapter 1) and the end of the Code (Chapters 18, 19, and Appendices) are always close together with an important contact zone. It appears that this general information should be grouped together, probably in the beginning of the Code.

Based on the modularity analysis, it seems that a reasonable proposal for the reader-friendly version of the Code would be the following: First Part: Introduction and Generalities, General Recommendations, Code of Ethics, Commission and Regulations; Second Part: General information on Publication, Availability, and Validity; Main Part: Species group, Genus group, Family Group. As all these conceptual modules are not independent but rather present extensive contact zones, it might be desirable that certain information from these contact zones should be repeated in various parts of the Code to provide both conceptual closure of the various concepts as well as connection between the different concepts.

A final point to be made refers to the Glossary. As expected, the Glossary is highly parcellated (0,890), and its terms are included in all modules and distributed quite evenly across modules. Currently, the Glossary is placed in the end of the Code, and both in print and online versions it requires extra effort to check the Glossary terms. In many cases, as exemplified in detail in my earlier paper (Vlachos, 2019) the understanding of the Glossary terms is key to understand and apply the Code. So, a final change to greatly improve the structure and readability of the Code would be to include all the relevant Glossary terms in the same printed or online page that they appear, perhaps in the form of footnotes or hove-over windows respectively, or any other technical solution that might fit better to that purpose.

The modularity analysis could help in providing a new visual representation and structure of the Code (Fig. 4). Each sub-module is conceptualized as a different “nomenclatural element”, and using their connections inside and outside their larger module, we can place them in a non-overlapping two-dimensional way, forming a table (Fig. 4). In this diagram, the basic Principles have, just like in the network, a central position. From left-to-right we go from more general elements to more specific ones, whereas the mid-and-right part of the diagram is divided in the four important structural items: availability, validity, formation of names, and typification. The right part of the diagram is divided vertically and successively from left-to-right in the species-, genus-, and family-group. Some modules are entirely included in one of these five structural areas, whereas others not. The various network metrics of each element could help in understanding the importance of each element (see (Vlachos, 2019) for detailed explanation of the various metrics). The number of items and average weighted degree tell the reader the amount and importance of the information included in this element. Elements with high density and high clustering are “heavy” elements, suggesting that these elements could stand and be understood on their own. Elements with low density and low clustering are more “unstable”, meaning that need to be read and understood in combination with other surrounding elements. This diagram could assist in using and teaching the Code, as well as to provide a starting point for the development of a virtual application.

Figure 4 The periodic table of nomenclatural elements.

The modularity analysis of the network of the Code allows dividing the Code into 65 different conceptual elements and place them in a non-overlapping, two-dimensional way. Each element includes its four main metrics. The most important elements have high number of items and high degree. “Heavy” elements are those with high density and clustering, and usually stand and be understood on their own, where “unstable” elements with low density and clustering need to be understood together with their surrounding elements.

Supplemental Information

Supplemental Information 1 The network with the various recovered modules

All original network files in the gephi format, with the recovered modules.

Click here for additional data file.

Supplemental Information 2 Information on the various modularity runs

This file contains that data of each modularity run for the entire network (first round) and for each module (second round), providing the modularity Q and number of modules, as well as the standard deviation of the runs. The selected one (highest Q) is indicated with boldface.

Click here for additional data file.

Supplemental Information 3 Click here for additional data file.

I would like to thank the Thomas Pape and the International Commission on Zoological Nomenclature for granting me permission to use the text of the Code for the analyses herein.

Additional Information and Declarations

Competing Interests

Author Contributions

Data Availability

The authors declare there are no competing interests.

Evangelos Vlachos conceived and designed the experiments, performed the experiments, analyzed the data, prepared figures and/or tables, authored or reviewed drafts of the paper, and approved the final draft.

The following information was supplied regarding data availability:

Raw data, including raw networks and the recovered modules and information on the various runs of the modularity analysis, are available in the Supplemental Files.

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
