# Peer review of "A modularity analysis helps improving the structure of the International Code of Zoological Nomenclature"

_PeerJ, doi:10.7717/peerj.10815_

## Round 0.1 · accepted · Accept

Thank-you for again submitting a manuscript to PeerJ. Based on the submitted reviews, I am happy to accept your submission for publication.

The review by Dr. Polaszek provided no critique of the manuscript other than questioning whether or not it would be of interest to readers. My view is that the review is not pertinent to a decision on acceptance or rejection of the manuscript.

·

Basic reporting

The article is logically organized and well-written. The information- and concept-dense text is made more accessible, and effectively summarized, by use of innovative graphics, such as the "periodic table" of elements of nomenclature.

Experimental design

The author makes clear the goals of this analysis of the content and organization of the Code, as well as the analytical methods used to assess both its current structure and the potential benefits of a different organization. A difficult, dense subject matter is presented very well by virtue of how the analysis was approached and presented.

Validity of the findings

The conclusions are stated well, as are the implications for the results of the study. Methods are made clear and data appropriately summarized.

Additional comments

This is the first comprehensive reexamination of the structure and conceptual content of the zoological code. The Code is understandably regarded as a "dry" subject; a bunch of rules, regulations, and recommendations that are necessary to maintain order in the ways in which species and higher taxa are named, but tolerated more than embraced as a conceptual framework for nomenclature and expression of taxonomic decisions. Examining the Code through the lens of information science creates an exciting opportunity to reimagine its organization and structure. As implied by the author, an "e-Code" in the future need not exist in a single, linear format, but could be accessed in more than one form by users. The manuscript poses an implicit challenge to the next international congress on zoological nomenclature to take a serious look at how the ideas and provisions of the Code are organized, thought about, and accessed by users. Negative aspects of the competing PhyloCode have been well documented and should have made it dead on arrival. The PhyloCode would arrogantly impose contemporary ideas about phylogenetic relationships upon future generations, undermining intellectual freedom and making names immutable monuments to the hubris of the current generation; in this ill-conceived PC code, names become conceptually rigid constructs, dramatically differing from the flexible names in the current Code that have proven their ability to keep pace with the growth of taxonomic thought. One (silly) reason the PhyloCode has any traction at all (and it should have none for anyone concerned about the testing and growth of scientific ideas about species and taxa, or with the humility to accept that we cannot foresee theoretical developments that may come about in the future) is the false perception that the existing Code, by virtue of its pre-phylogenetic origins and age, is somehow not up to meeting contemporary needs. In addition, the Code looks cobbled together because it has grown in fits and starts through periodic modifications and edits. This paper approaches the Code and its concepts objectively, with fresh eyes, and offers an interesting opportunity to make it more purposefully structured, accessible and flexibly organized to meet user needs. The Code serves zoology extremely well and should be tweaked, improved and built upon to meet community needs, not replaced. This manuscript prescribes one way to approach this challenge. Impacts of this paper may well be to shape the agenda of the next congress, to reimagine how the Code is presented and used in electronic form, and to change the perception of the Code by seeing it as a dynamic network of ideas for organizing knowledge of biological diversity. The Commission boldly embraced electronic publication of new species and nomenclatural acts; it is time to think of the Code itself in a more flexibly organized electronic form. I am not yet certain of all the implications of this analysis of the conceptual content of the Code, or the extent to which I agree with its conclusions, but I am sure that the time has come to breath new life into the Code by making it more easily accessed and understood.

·

Basic reporting

There are quite a number of errors in the use of English throughout the text.

Experimental design

I am sure the author has conducted the analysis correctly.

I am much less certain that the article is of any interest to readers of PeerJ. While the Code is definitely in need of reform, I am not sure that this article, published publicly, will help that reform process significantly (see below).

Validity of the findings

As stated above, I am not convinced that the publication of this article will help the process of reforming the Code. I think that such a specialised analysis could be presented to the writers of the next version of the Code, but I fail to see that it could be of any interest to anyone else - I may be wrong of course.

The analyses of the 9 "modules" is unlikely to be of interest to more than a small number of individuals globally, and the general recommendations could be presented directly to the Commission, who I am sure would take them into consideration when amending or re-writing the Code.

Additional comments

I am not convinced that the publication of this article will help the process of reforming the Code. I think that such a specialised analysis could be presented to the writers of the next version of the Code, but I fail to see that it could be of any interest to anyone else - I may be wrong of course.

The analyses of the 9 "modules" is unlikely to be of interest to more than a small number of individuals globally, and the general recommendations could be presented directly to the Commission, who I am sure would take them into consideration when amending or re-writing the Code.

·

Basic reporting

Well-written, clearly presented.

Experimental design

Good design, interesting methods.

Validity of the findings

Conclusions well stated, looking forward to test some of these hypotheses and implement the suggested changes in the Code.

Additional comments

Work towards the Fifth Edition of the International Code of Zoological Nomenclature by is currently taking place. The information submitted for publication by Dr. Vlachos is well-written and provides a different approach to look at, and improve, the contents of the current edition of the Code. As a member of the editorial committee working on the Fifth Edition of the Code, I will be looking forward to using data in the published version of this manuscript to test some of the hypotheses proposed therein. Note that there will be a period for community input regarding suggested changes by our editorial committee and it will be important to hear from all users regarding how the next edition of the Code could be structured in order to enhance comprehension and usage for all users (including students and beginners). Overall a very positive contribution.